# The concept of "whole perforator system" in the lateral thoracic region for latissimus dorsi muscle-preserving large flaps: An anatomical study and case series

**Yu Kagaya**[¤a]*, **Masaki Arikawa, Takuya Sekiyama, Hideyuki Mitsuwa, Ryo Takanashi, Marie Taga, Satoshi Akazawa, Shimpei Miyamoto**[¤b]

Department of Plastic and Reconstructive Surgery, National Cancer Center Hospital, Tsukiji, Tokyo, Japan

¤a Current address: Department of Plastic and Reconstructive Surgery, Kyorin University Hospital, Mitaka, Tokyo, Japan
¤b Current address: Department of Plastic and Reconstructive Surgery, The University of Tokyo Hospital, Hongo, Tokyo, Japan
* mkagayakson@yahoo.co.jp

**Data Availability Statement:** All relevant data are within the manuscript and its Supporting Information files.

## Abstract

### Background

Previous studies have reported on the abundant cutaneous perforating blood vessels around the latissimus dorsi (LD) lateral border, such as a thoracodorsal artery perforator (TDAP) of septocutaneous type (TDAP-sc) and muscle-perforating type (TDAP-mp), or the lateral thoracic artery perforator (LTAP). These perforators have been clinically utilized for flap elevation; however, there have been few studies that accurately examined all the cutaneous perforators (TDAP-sc, TDAP-mp, LTAP) around the LD lateral border. Here, we propose a new "whole perforator system" (WPS) concept in the lateral thoracic region and a methodology that enables elevating large flaps with reliable perfusion in a muscle-preserving manner.

### Methods

We first performed an anatomical study that verified the number and perforating points of all perforators around the LD lateral border using the results of dynamic contrast-enhanced magnetic resonance imaging of patients with breast cancer. Following the anatomical evaluation, we performed large muscle-preserving flap transfer that contained all of the perforators around the LD lateral border in an actual clinical setting.

### Results

A total of 175 latissimus dorsi from 98 patients were included. The mean number of perforators (TDAP-sc + TDAP-mp + LTAP) per side was 4.51±1.44 (2–9); TDAP-sc was present in 57.1% (100/175) of cases, and TDAP-mp in 76.6% (134/175); the TDAP total prevalence rate (TDAP-sc + TDAP-mp) was 96.0% (168/175). The LTAP existence rate was 94.3%

**Funding:** The authors received no specific funding for this work.

**Competing interests:** The authors have declared that no competing interests exist.

(165/175). Distance from the axillary artery to the TDAP-sc was 148.7±56.3 mm, which was significantly proximal to the TDAP-mp (183.8±54.2 mm) and LTAP (172.2±81.3 mm).

## Conclusion

The lateral thoracic region has an abundant cutaneous perforator system derived from the descending branch of the thoracodorsal and lateral thoracic arteries. Clinical application of the lateral thoracic WPS flap is promising, with a large survival area even with muscle-preserving procedures and requiring a relatively simple procedure.

## Introduction

The thoracodorsal artery perforator (TDAP) flap has been widely utilized as a soft, thin perforator flap with reliable perfusion since its conception by Angrigiani et al. [1] in 1995. The TDAP flap usually utilizes muscle-perforating perforators (TDAP-mp) [2], which are frequently found around the lateral border of the Latissimus dorsi (LD) muscle [1–4]. In addition, a perforator named TDAP-sc (septocutaneous type) has been acknowledged and utilized for flap elevation, which derives from the thoracodorsal artery (TDA) but runs an anterior circum-muscular course and does not perforate the LD muscle [5–7]. Moreover, the lateral thoracic artery (LTA), which usually runs longitudinally anterior to the LD muscle, and its cutaneous perforator of the lateral thoracic artery perforator (LTAP) have also been acknowledged as a flap pedicle of the adjacent area with a TDAP flap [8, 9].

Thus, there are abundant cutaneous perforators around the LD lateral border that can be utilized for flap elevation. Several studies have examined the number or perforating point of TDAPs or LTAPs in clinical cases [2], imaging studies [4], or cadaveric studies [5, 6]. Further studies have reported on the actual clinical utility of these perforators for flap elevation [7–12]; however, no studies have comprehensively examined the prevalence rate and number of all the cutaneous perforators (TDAP-sc, TDAP-mp, LTAP) around the LD lateral border.

A latissimus dorsi musculocutaneous (LDMC) flap contains multiple TDAPs and can perfuse a larger skin paddle than a usual TDAP flap [13, 14]; however, the bulky muscle may prove to be problematic, and donor site morbidity due to removing the LD muscle is not negligible [15, 16]. There are cases that require thin and large thoracodorsal area flaps, in which case, it is difficult to choose between TDAP or LDMC flaps as the former can achieve a "thin" flap; however, a variation of perforator sizes and locations is of concern for the perfusion area [2, 17, 18], and an LDMC flap can achieve a "large" flap, but the bulkiness of the LD muscle brings an excessive flap volume. To solve this problem, we hypothesized that a flap with a large skin paddle without the LD muscle can be elevated by containing all the cutaneous perforators (TDAP-sc, TDAP-mp, LTA), existing around the LD lateral border.

Based on the above hypothesis, we first verified the number and perforating points of all perforators (TDAP-sc, TDAP-mp, LTAP) of the area around the LD lateral border using the results from the dynamic contrast-enhanced magnetic resonance imaging (DCE-MRI) of patients with breast cancer. Breast DCE-MRI is considered suitable for examining vessels around the LD lateral border because clinical DCE-MRI breast examinations are commonly performed with fat suppression and are optimized for evaluating tumor vascularity, providing high contrast between contrast agents and muscles or suppressed fat [19–21]. Moreover, the area of the LD lateral border is automatically included in the usual target range.

After anatomical evaluation, by utilizing the insights provided by the imaging study, we performed large muscle-preserving flap transfers in an actual clinical setting, which contained

all the perforators around the LD lateral border. The results of the flap transfer were successful and promising; therefore, we report both the results of anatomical examination as background evidence and the results of the case series as clinical applications. Additionally, we propose a new "whole perforator system" (WPS) concept in the lateral thoracic region and a methodology that enables elevating large flaps with reliable perfusion, with a simple procedure in a muscle-preserving manner.

## Materials and methods

The present article was written according to the "strengthening the reporting of observational studies in epidemiology" statement [22, 23]. This study was approved by the Institutional Review Board of National Cancer Center Japan and was conducted in accordance with the Declaration of Helsinki on investigation in humans. Although informed consent was obtained in the form of an opt-out option on the website, for the three patients whose photographs were used in the present article, informed consent was obtained from each patient on an individual basis.

We conducted a retrospective review of the medical records of breast cancer patients who underwent primary breast mastectomy between January and December 2018 at the National Cancer Center Hospital, Japan and were consulted by the Department of Plastic and Reconstructive Surgery. Preoperative breast DCE-MRI was utilized for perforator examination around the LD lateral border. The MRI apparatus was a 3.0 Tesla whole-body magnetic resonance system (Achieva 3.0T X-series®, Philips Healthcare, Netherlands) with a dStream Breast 16-channel coil. Dynamic contrast enhancement was performed by injecting 0.1 mmol/kg of gadobutrol (Gadovist IV Inj.1.0mol/L Syringe®, Bayer Yakuhin, Ltd., Japan) at a rate of 1.0 mL/s via an automatic injector, followed by a 20 mL flush of saline. The imaging protocol utilized for perforator evaluation was an enhanced-T1 high-resolution isotropic volume examination (eTHRIVE) (Philips Healthcare, Netherlands) in an early phase (60–150 seconds) with a 1.6-mm slice.

All perforators on the LD muscle within 3 cm from the lateral border and anterior to the LD up to the pectoralis major muscle (Fig 1) were examined in terms of number and location, measured as the distance from the axillary artery. The types of perforators that existed in the area were TDAP-sc, TDAP-mp, and LTAP. All the evaluated TDAPs were derived from the descending branch of the TDA because of the target area for evaluation. The criteria to be identified as a valid perforator in the present study were "a clearly recognizable branching point from the main arteries (TDA or LTA) and a running course that is continuously perforating toward the subcutaneous tissue (Figs 1 and 2)." Therefore, small or unclear perforators were ignored under the criteria. By implementing this exclusion, clinically available and usable perforators around the LD lateral border were counted in the present study. In addition, regarding LTAP, a perforator had to go over-across the LD lateral border". Moreover, the number of confluences of TDA and LTA or TDAP and LTAP were counted as reference data.

### Statistical analysis

The Mann-Whitney U test was used to analyze differences between groups. All analyses were performed using the "R" software program (version 4.0.2). P values <0.05 were considered to indicate statistical significance.

### Results

Data of 156 consecutive patients were extracted in the predetermined period. Patients who did not undergo preoperative DCE-MRI or sides (right or left) that had images of unclear

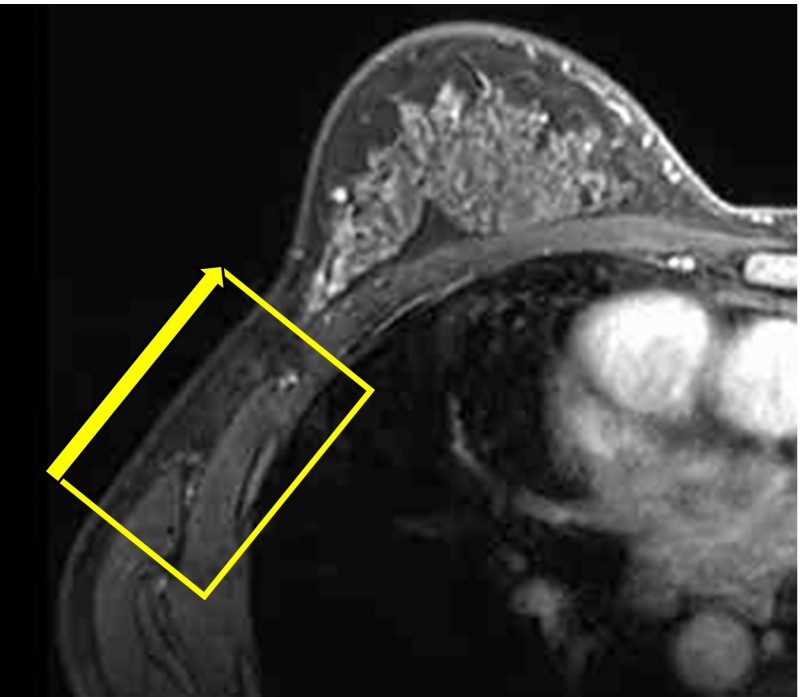

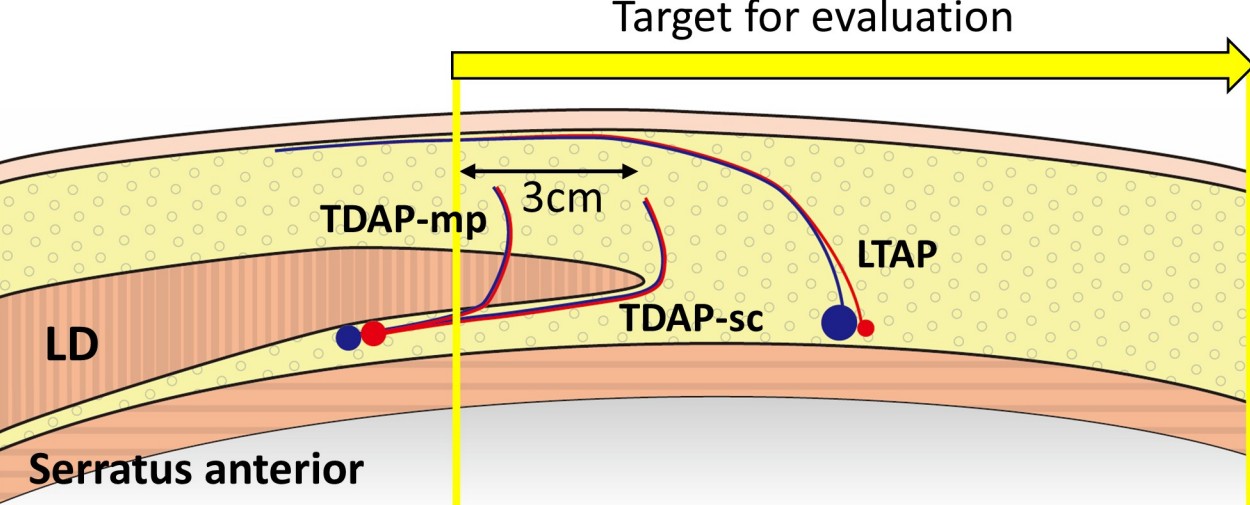

**Fig 1. Target area for perforator evaluation.** All the perforators on the LD muscle within 3 cm from the lateral border and anterior to the LD up to the pectoralis major muscle were evaluated.

resolution due to artifacts or an incomplete target area due to a large body frame were excluded. Finally, 175 sides of 98 patients were included in the evaluation. For the eligible cases, the mean age was 48.4±8.7 (28–78) years, and 93 sides were diseased (breast cancer), while 82 were healthy. The overall results of the perforator evaluation are shown in Table 1. The mean number of perforators (TDAP-sc + TDAP-mp + LTAP) per side was 4.51±1.44 (2–9); TDAP-sc existed in 57.1% (100/175), and TDAP-mp was in 76.6% (134/175); the TDAP total prevalence rate (TDAP-sc + TDAP-mp) was 96.0% (168/175). The LTAP prevalence rate was 94.3% (165/175). Considering the possibility of differences in perforator development

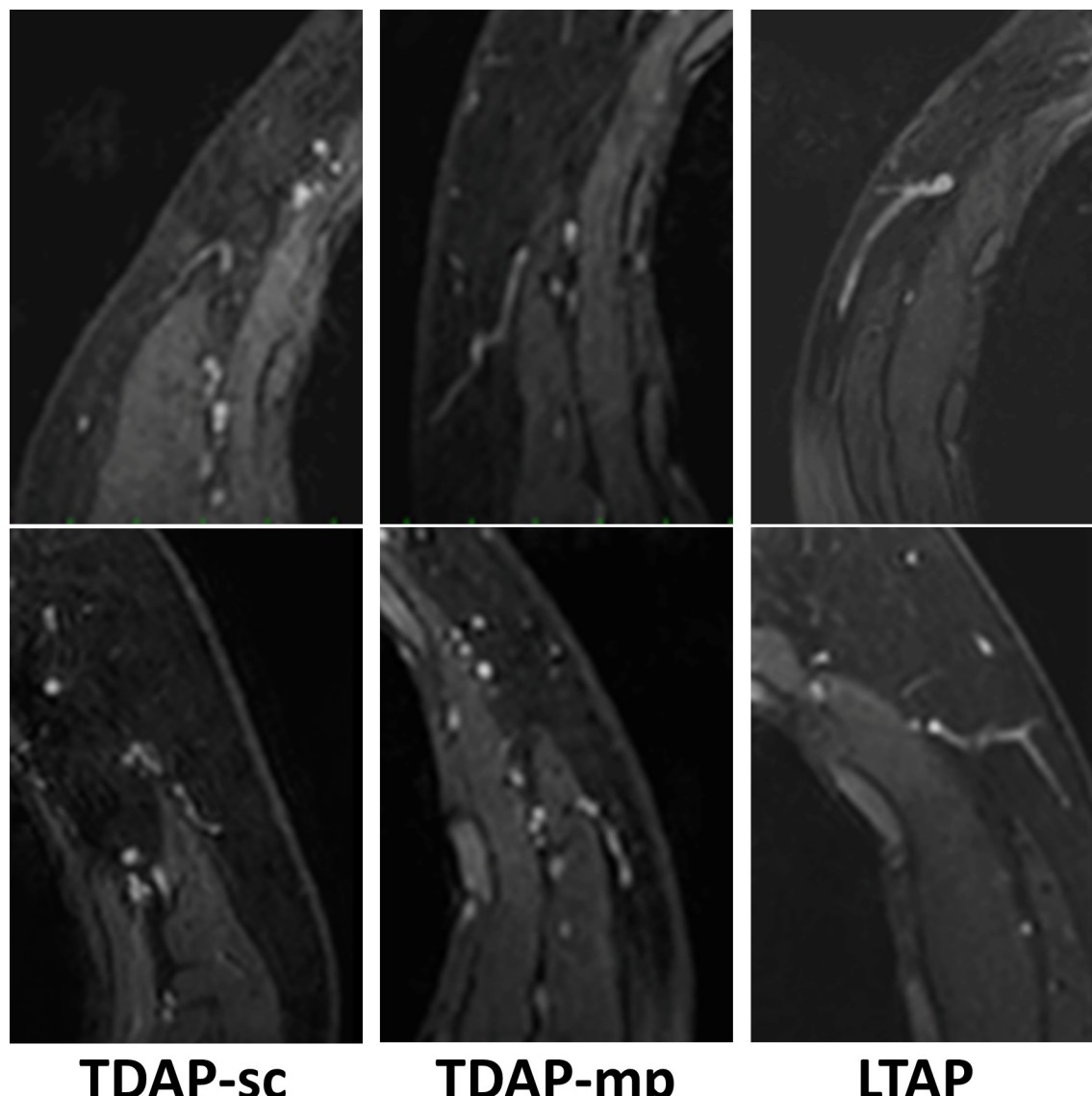

**TDAP-sc**     **TDAP-mp**     **LTAP**

**Fig 2. Typical appearances of perforators (TDAP-sc/TDAP-mp/LTAP) in the DCE-MRI.** The TDAP-sc runs a circum-muscular course, TDAP-mp runs a muscle-perforating course, and LTAP runs over-across the LD muscle.

between the diseased and healthy sides, a perforator number comparison between the two groups was conducted (Table 2). Consequently, there was no significant difference in perforator number in all categories, so the results of the present study were considered not to be affected by the presence of breast cancer.

**Table 1. A data summary of perforators on all the 175 sides.**

|  | TDAP-sc | TDAP-mp | TDAP total (sc+mp) | LTAP | Total (TDAP+LTAP) |
|---|---|---|---|---|---|
| **Prevalence rate** | 57.1% (100/175) | 76.6% (134/175) | 96.0% (168/175) | 94.3% (165/175) | 100.0% (175/175) |
| **Perforator number** | 0.92±1.06 (0~5) | 1.24±0.92 (0~4) | 2.16±1.17 (0~7) | 2.35±1.16 (0~6) | 4.51±1.44 (2~94) |
| **Distance from the axillary artery** | 148.7±56.3mm (8~272) | 183.8±54.2mm (48~352) | 168.8±73.1mm (8~352) | 172.2±81.3mm (8~336) | 170.6±71.0mm (8~352) |

TDAP-sc: Thoracodorsal artery perforator septocutaneous type. TDAP-mp: Thoracodorsal artery perforator muscle-perforating type. LTAP: Lateral thoracic artery perforator.

**Table 2.  Comparison of perforator number between the diseased and healthy sides.**

|  | Diseased side (n = 93) | Healthy side (n = 82) | P value* |
|---|---|---|---|
| **Overall (TDAP-sc/-mp +LTAP)** | 4.61±1.52 (2–9) | 4.40±1.35 (2–9) | 0.38 |
| **TDAP-sc** | 0.99±1.15 (0–5) | 0.84±0.95 (0–4) | 0.55 |
| **TDAP-mp** | 1.22±0.88 (0–4) | 1.27±0.97 (0–4) | 0.77 |
| **TDAP overall (sc+mp)** | 2.20±1.26 (0–7) | 2.11±1.05 (0–5) | 0.99 |
| **LTAP** | 2.41±1.17 (0–5) | 2.29±1.16 (0–6) | 0.62 |

* Mann-Whitney U test. TDAP-sc: Thoracodorsal artery perforator septocutaneous type. TDAP-mp: Thoracodorsal artery perforator muscle-perforating type. LTAP: Lateral thoracic artery perforator.

Distance from the axillary artery to the TDAP-sc was 148.7±56.3 (8–272) mm, to the TDAP-mp 183.8±54.2 (48–352) mm, and to the LTAP 172.2±81.3 (8–336) mm. TDAP-sc was revealed to be significantly proximal to the TDAP-mp and the LTAP (both P<0.01), although there was no significant difference in the location between TDAP-mp and LTAP.

The confluence of TDA and LTA (Fig 3) was confirmed in 28.0% (49/175) of the cases, and the confluence of TDAP and LTAP (Fig 3) was confirmed in 28.6% (50/175) of the cases. In total, either of the confluences was confirmed in 51.4% (90/175) of the patients.

## Clinical applications (Table 3, Figs 4A, 4B, 5A, 5B)

The results of the perforator evaluation revealed that multiple reliable perforators based on the descending branch of TDA and LTA exist around the LD lateral border. Therefore, we launched a clinical application for a flap, which includes all perforators around the LD lateral border. We named this flap as the "whole perforator system (WPS)" in the lateral thoracic area. The lateral thoracic WPS flaps were elevated and transferred to the defect after extensive sarcoma resections in seven cases from June 2019 to March 2020.

The summarization of the flap elevation was as follows: 1: anterior skin incision far from the LD lateral border; 2: dissection just above the serratus anterior muscle toward under the LD, automatically including all TDAPs and LTAPs, both valid and invalid according to the inclusion criteria previously described in the Methods section; and 3: selection of complete muscle-preserving or muscle-sparing flaps according to the preoperative imaging study (Figs 6 and 7). Preoperative DCE-MRI is not indispensable; however, acknowledging the location of dominant perforators by Doppler ultrasonography is useful for selecting the flap type (complete muscle-preserving or muscle-sparing), although each perforator is not actually detected or dissected in the current technique. The width of the muscle attached to the flaps was within 3 cm from the LD lateral border. The root of the flap pedicles was TDA with or without LTA, and thoracodorsal vein (TDV) with or without lateral thoracic vein (LTV). The thoracodorsal nerve was preserved in all cases. Detailed data of the seven consecutive cases are shown in Table 3. There were five free and two pedicled flaps, with a mean flap size of 12.2 cm × 27.0 cm (width × length). The flaps completely survived in all but one case of distal partial necrosis.

## Discussion

The present study evaluated perforators around the LD lateral border (TDAP-sc/TDAP-mp/LTAP) using the results of DCE-MRI from patients with breast cancer. The mean number of perforators (TDAP-sc + TDAP-mp + LTAP) was 4.51±1.44. The TDAP total prevalence rate (TDAP-sc + TDAP-mp) was 96.0% (TDAP-sc: 57.1%, TDAP-mp: 76.6%) and that of LTAP was 94.3%. Based on these results, we performed a clinical application for a flap that includes

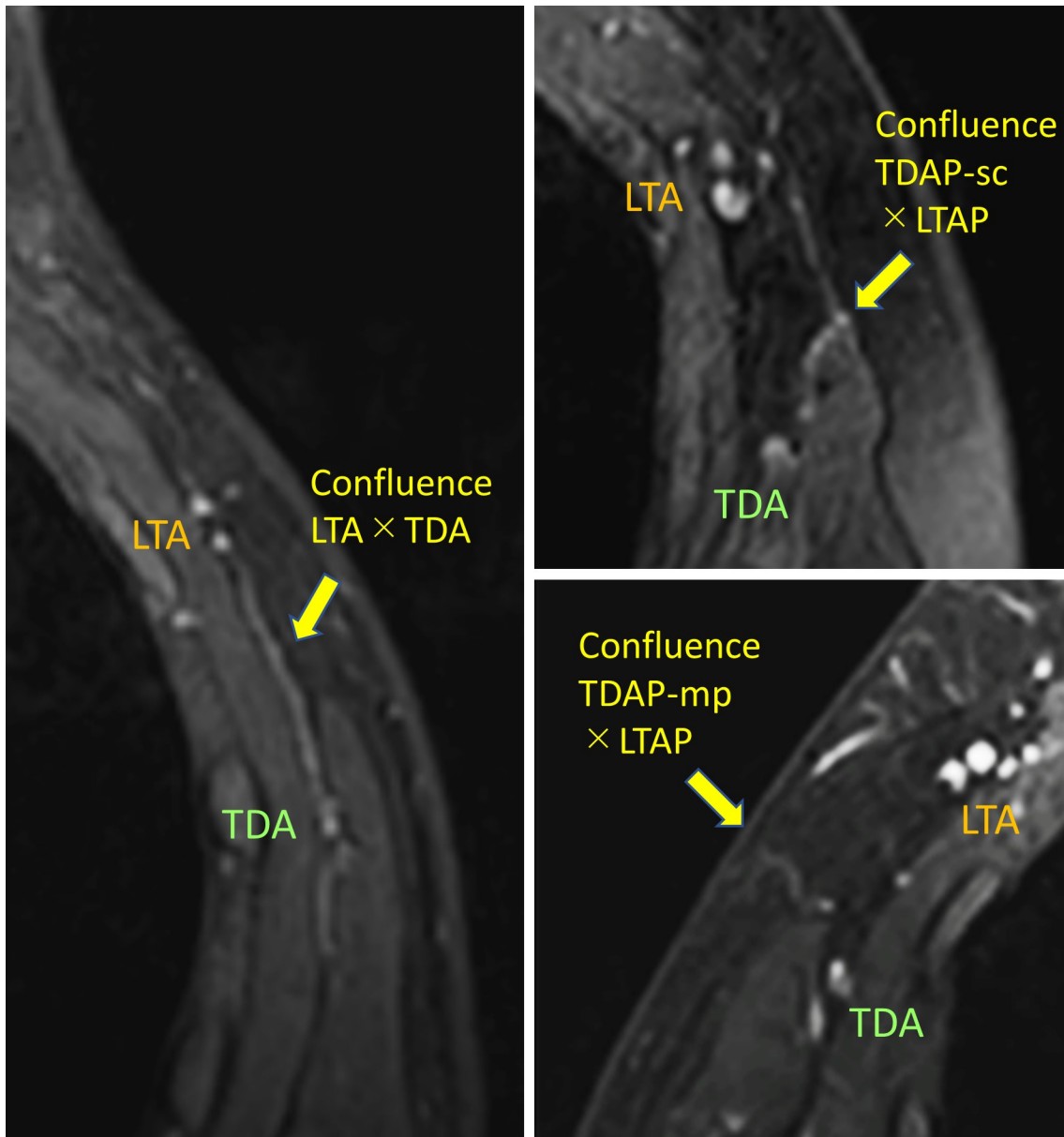

**Fig 3. Typical examples of confluence of TDA/LTA and TDAP/LTAP.** The image of left side shows a communication of TDA/LTA (relatively distal portion), and the right side shows communications of cutaneous perforators (TDAP and LTAP).

the whole of the perforators (WPS) in the lateral thoracic area and succeeded in seven large flap transfers in a muscle-preserving manner.

The present study revealed that there were many cutaneous perforators around the LD lateral border, though they did not necessarily perforate the LD muscle. Generally, when elevating a traditional LDMC flap, perforators anterior to the LD are naturally ligated/coagulated or not included in the flap before identifying the lateral border of the LD (Fig 8) [24]. The anatomy and prevalence of perforators around the LD lateral border vary between cases; therefore, if the TDAP-sc or LTAP, which do not perforate the LD muscle, is the dominant perforator in charge of skin perfusion in the thoracodorsal area, the survival area of the traditional LDMC flap can be small due to the abandonment of dominant perforators.

**Table 3. Detailed data of the seven consecutive cases in which the lateral thoracic "whole perforator system" flap was utilized.**

| | Age Sex | Region | F/P | Flap size (cm×cm) | Flap type | Pedicle Artery | Pedicle Vein | Donor-site closure | Flap survival | Postoperative complication (Clavien-Dindo Classification) |
|---|---|---|---|---|---|---|---|---|---|---|
| 1 | 47 F | Lower leg | F | 28×16 | MS | TDA | TDV | skin graft | complete survival | IIIa (flap repositioning for the release of tension on the vascular pedicle) |
| 2 | 76 M | Thigh | F | 25×16 | CP | TDA/LTA | TDV/LTV | skin graft | complete survival | not applicable |
| 3 | 84 M | Upper arm | P | 22×10 | CP | TDA/LTA | TDV/LTV | primary closure | complete survival | not applicable |
| 4 | 82 M | But-tock | F | 30×10 | CP | TDA | TDV/LTV | primary closure | partial necrosis (distal tip) | I (removal of stitches at the bedside for the release of flap congestion) |
| 5 | 41 M | Thigh | F | 23×9.5 | MS | TDA | TDV/LTV | primary closure | complete survival | not applicable |
| 6 | 59 M | Thigh | F | 29×14 | MS | TDA | TDV/LTV | primary closure (rotation flap) | complete survival | not applicable |
| 7 | 62 M | Shoul-der | P | 32×10 | MS | TDA | TDV | primary closure | complete survival | not applicable |

F/P: Free flap or pedicled flap. Flap type: Muscle-sparing type (MS) or Complete muscle preservation type (CP). TDA/TDV: Thoracodorsal artery and vein. LTA/LTV: Lateral thoracic artery and vein.

All cutaneous perforators around the LD lateral border exist in the subcutaneous layer; therefore, dissection just above the serratus anterior muscle from the anterior skin incision automatically includes all TDAPs in subcutaneous fat; the flap can be elevated as an adipose-cutaneous flap (complete muscle preservation) (Fig 6) or muscle-sparing LDMC flap (Fig 7) according to perforator types. In this way, the sole perforator dissection is not needed for flap elevation, and a large LD muscle is unnecessary for skin flap perfusion. There can be many capillary perforators around the LD lateral border, which was too small to be enhanced and visible on DCE-MRI. Koshima [25] reported a TDAP flap that included only capillary perforators. If the flap is elevated in the above-mentioned manner, both visible and invisible perforators can be included in the flap. A lateral thoracic region adipose-cutaneous flap with a similar concept of including TDAP-sc and LTAP has been reported before [26], although it was utilized as a local flap. It is considered that a visible and reliable mean of 4.51 perforators and many invisible capillary perforators can secure a substantial flap survival area without attachment to the large LD muscle.

The range of perforator evaluation (within 3 cm from the lateral border of the LD) was determined based on past reports and the actual convenience of flap elevation. Several studies have reported that most TDAPs are derived from the descending branch of the TDA, located within 3 cm from the LD lateral border [2, 4, 5, 14]. In terms of operative procedure, the LD lateral border is thin so that it is easy to cut and attach a 3-cm wide LD muscle to the flap. Regarding TDAP-sc, while it has been acknowledged as a direct cutaneous perforator of TDA [5, 6], the clinical application for flap elevation has been reported just recently [7]. Previous cadaveric studies reported that the existence rate of TDAP-sc was 55% [5] or 60% [6], which is quite similar to the result of the present study (57.1%). In addition, the short pedicle length of TDAP-sc was reported as one of the shortcomings of the perforator [2], which coincided with the results of the present study (TDAP-sc located significantly proximal to TDAP-mp and LTAP). However, if the existence of a dominant TDAP-sc is confirmed by preoperative imaging tests, TDAP-sc should be included in the lateral thoracic WPS flap in order to obtain better perfusion.

There have been several reports regarding flaps utilizing a sole LTAP, which mentioned that the perfusion area of the LTAP was located more anterior to that of the TDA [8, 9, 17, 27]. In addition, the LTA and LTV are anatomically complicated and have much variation, and

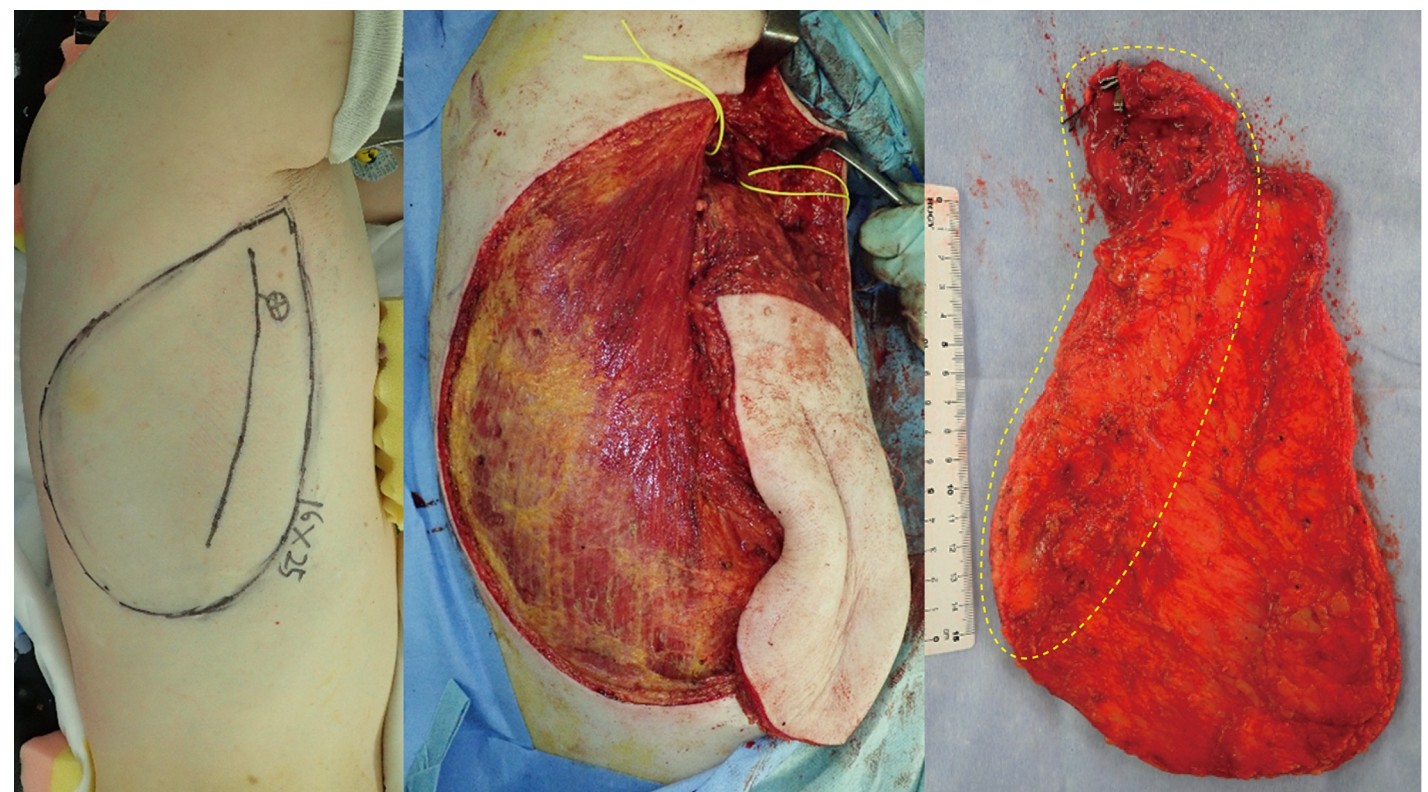

**Fig 4.** A, B. Case no. 2 in Table 3. Reconstruction after extensive resection of the myxofibrosarcoma of the proximal thigh. Preoperative color Doppler ultrasonography shows a dominant TDAP-sc, such that the flap was elevated as a 25×16 cm adipose-cutaneous flap, with complete LD preservation (Fig 4A, yellow dotted area shows the tissue including "WPS"). As a vascular pedicle, the TDA and LTA formed a confluence, and a single arterial anastomosis was performed utilizing the confluence (Fig 4B). The TDV and LTV also formed a confluence; however, the confluent section was short and contained a venous valve, so that the pedicle veins were divided, and two venous anastomoses were performed (Fig 4B). The postoperative course was uneventful, and the flap survived completely. The donor site was closed with a split-thickness skin graft (S1 Fig, 4 months postoperatively).

they usually run separately in the proximal region; however, distally, they run together at the perforator level [8]. Considering the variety of vascular anatomy of the lateral thoracic area [28, 29], some cases can have non-dominant TDAPs and complementarily dominant LTAPs that nourish the skin area more posterior than usual. There were seven sides among the 175 (4.0%) of no TDAP in the present study, which inevitably had dominant LTAPs. By combining all these findings, there is a high variance of perforators that nourish the skin of the lateral thoracic area between cases, so that TDAP-sc or LTAP, which are not utilized for usual TDAP flaps, are indispensable for skin flap perfusion and TDAP-mp. Therefore, the anterior skin incision should be located far anterior from the LD muscle to include all the perforator system (WPS) in the flap.

The procedure of the lateral thoracic WPS flap elevation is quite different from that of the traditional LDMC flaps, and several advantages and disadvantages specific to this flap are recognized and noted below. The advantages of the flap are that (1) complicated techniques are not needed as there is no sole perforator dissection, (2) the double pedicle vein (TDV and LTV) can be secured (double pedicle artery can also be dissected, but the root of the LTA is usually small), and (3) the donor site is anterior to that of the usual LDMC flap, so that the skin is softer and more elastic, being advantageous for primary closure. Moreover, the donor scar is considered more inconspicuous from the back in comparison to a traditional LDMC flap. Even the scar of skin graft can be hidden—to some extent—in the back view. The disadvantages of the flap are that (1) the vascular pedicle tends to be short and bulky because the adipose tissue including all proximal perforators is attached around the vascular pedicle; (2) individual differences in perforator distribution are significant, and preoperative imaging tests are desirable, especially for complete muscle-preserving type flaps (enhanced CT or MRI is not always necessary; however, preoperative Doppler ultrasonography is preferable for detecting the location of dominant perforators); and (3) donor-site morbidity; primary closure of the donor site can cause breast deformity (lateral traction), and there are concerns about long thoracic nerve injury. Dissection under the fascia of serratus anterior muscle can naturally cause long thoracic nerve injury, which can cause denervation of the serratus anterior muscle. However, no clinically problematic signs of shoulder movement disorder occurred in the present cases. The root of the long thoracic nerve is distant from the root of the flap pedicle; thus, the proximal part of the long thoracic nerve was preserved and the denervation of serratus anterior muscle was considered partial. However, surgeons should keep complications in mind when elevating this flap.

Regarding confluences of TDA and LTA (or TDAP and LTAP), more than half of the sides had findings of confluence in the enhanced MRI in the present study. There were many variations in the location of confluence, such as the root of the pedicle, before the branching point of the perforator or the perforator level in subcutaneous tissue (Fig 3). Previous cadaveric studies reported that 9.2% [29] or 16.8% [28] of cases had confluence of TDA and LTA at the root level. The present study had a higher prevalence of confluence than previous studies because all the confluences were counted, which were not limited to the root of the pedicle as mentioned above. In addition, it is possible that another invisible communication of the two vessels (TDA and LTA) at the capillary level existed. These confluences can be all included in the flap

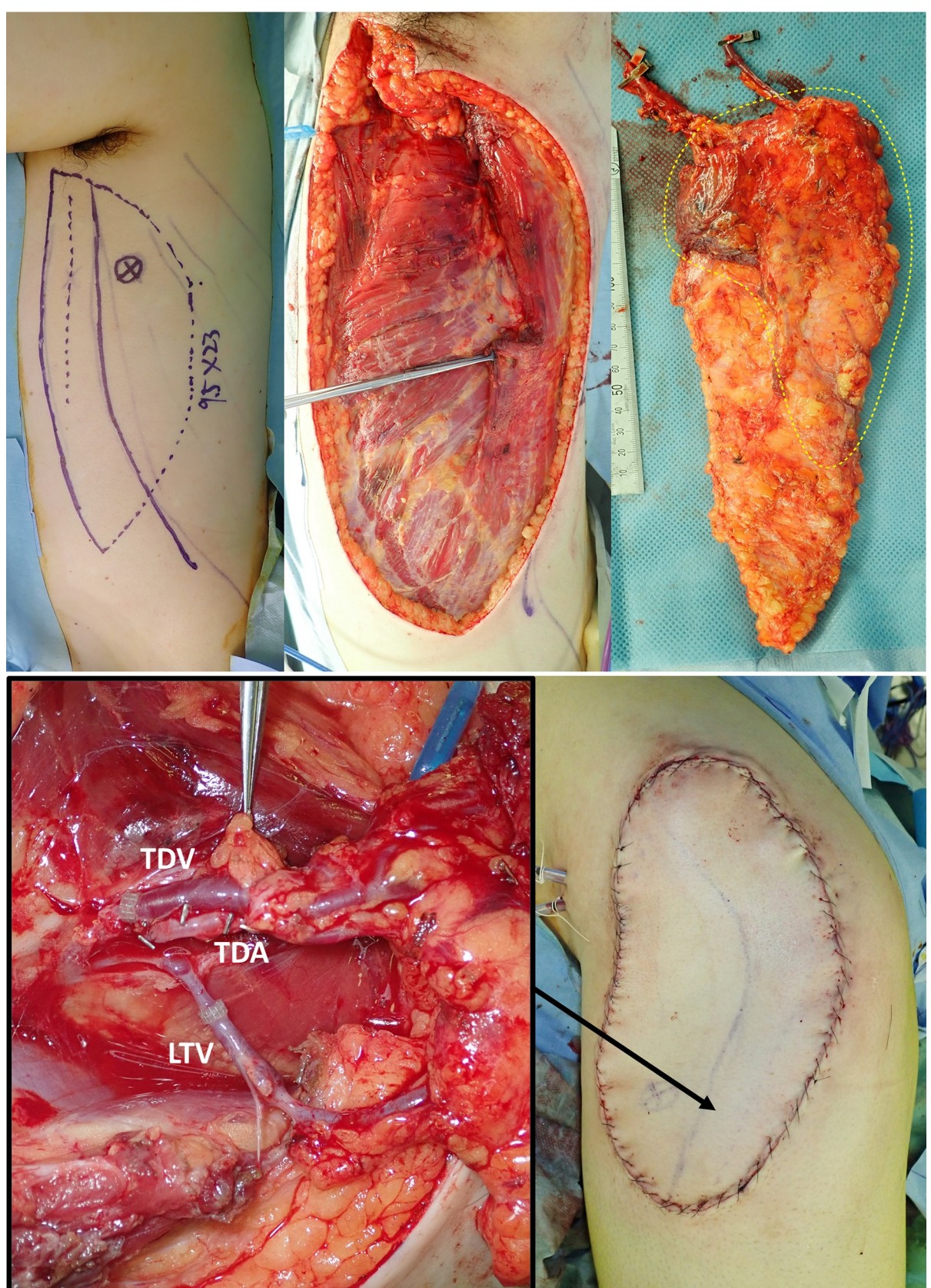

**Fig 5.** A, B. Case no. 5 in Table 3. Reconstruction after the extensive resection of the undifferentiated pleomorphic sarcoma of the proximal thigh. Preoperative color Doppler ultrasonography shows a dominant TDAP-mp, such that the flap was elevated as a 23×9.5 cm muscle-sparing LDMC flap (Fig 5A). The "WPS" in the adipose tissue anterior to the LD was all included in the flap, and the lateral border of the LD around the dominant TDAP-mp was attached little to the flap (Fig 5A, yellow dotted area shows the tissue including "WPS"). The vascular pedicles for anastomosis were the TDA and TDV/LTV (Fig 5B). The postoperative course was uneventful, and the flap survived completely. The donor site was closed primarily (S2 Fig, 6 months postoperatively).

if the above-mentioned method of the lateral thoracic WPS flap elevation is performed, stabilizing flap perfusion.

We experienced one partial necrosis (venous congestion) of the distal tip among seven clinical cases of the lateral thoracic WPS flap (Fig 9). In this case, we attempted to have a long flap length and designed the flap in an oblique direction from the axillary area to the distal central back region. The lateral thoracic WPS flap was based on the perforators anterior to the LD and perforators derived from the descending TDA branch, so that the flap design should be parallel to the LD lateral border. The best perfusion area of the flap is considered around the average point of the perforators (Table 1, 14–19 cm from the axillary artery). Considering the results of imaging studies and clinical applications and past studies [14, 17, 30], the proximal edge of the flap can be stretched until the axillary area and the distal edge can be extended to the iliac crest if the flap is designed parallel to the LD lateral border.

Several limitations associated with this study need to be acknowledged. First, there was a bias in the study population as it included only women with breast cancer. The possible influence due to the subjects' sex is unknown, but it is possible that a cancer induced neovascularization [31–33], leading to the development of perforators, which could affect the results. However, the comparison between healthy and diseased sides showed no difference in the perforators' number per side, suggesting that the presence of cancer did not affect the results.

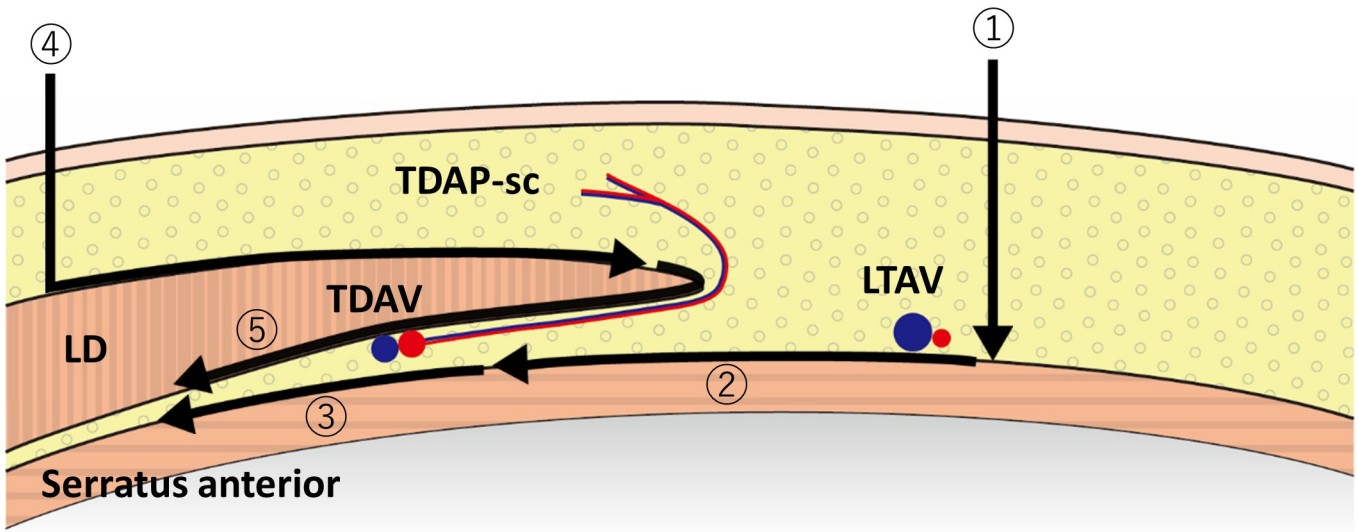

# Whole perforator system flap
## Complete muscle-preservation type

**Fig 6. A schematic illustration of an operational process of a lateral thoracic "WPS" flap (complete muscle preservation type).** ① Anterior skin incision far from the lateral border of the LD, ②③ dissection just above the serratus anterior muscle toward under the LD, automatically including all TDAPs and LTAPs, ④ posterior incision and dissection directly above the LD, and ⑤ dissection along the back side of the LD preserving the TDA/TDV and TDAP-sc.

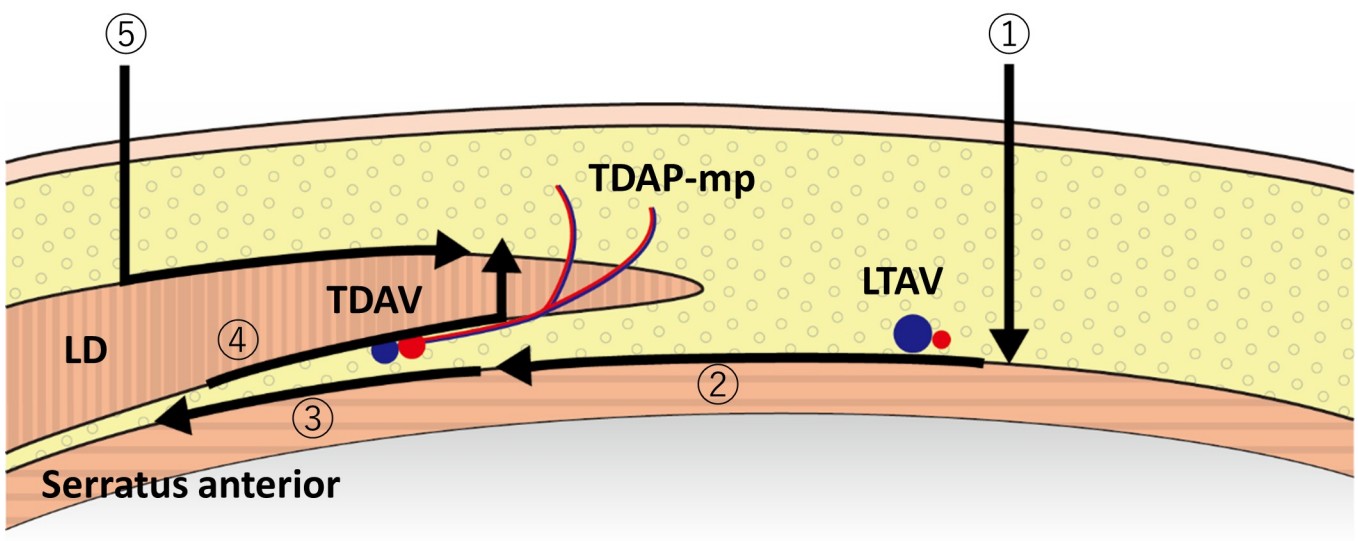

**Fig 7. A schematic illustration of an operational process of a lateral thoracic "WPS" flap (muscle-sparing type).** ①~③ The same procedure as the complete muscle preservation type (Fig 6), ④ dissection along the back side of the LD muscle and cutting the LD muscle medial to the dominant TDAP-mp, and ⑤ posterior incision and dissection just above the LD muscle.

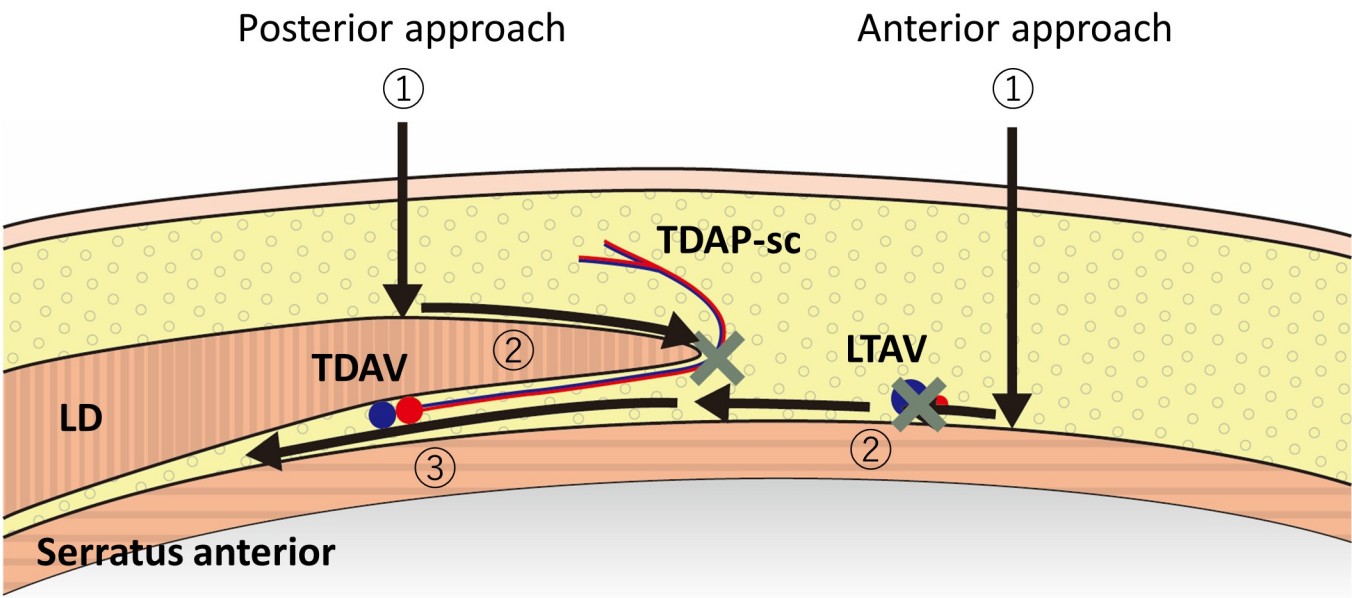

**Fig 8. A schematic illustration of an operational process of a traditional LDMC flap.** ①② At either anterior or posterior approaches, the first step is to identify the LD lateral border, in which the TDAP-sc or LTAP are naturally ligated/coagulated or not included in the flap, and ③ dissection under the LD and identifying the TDAV.

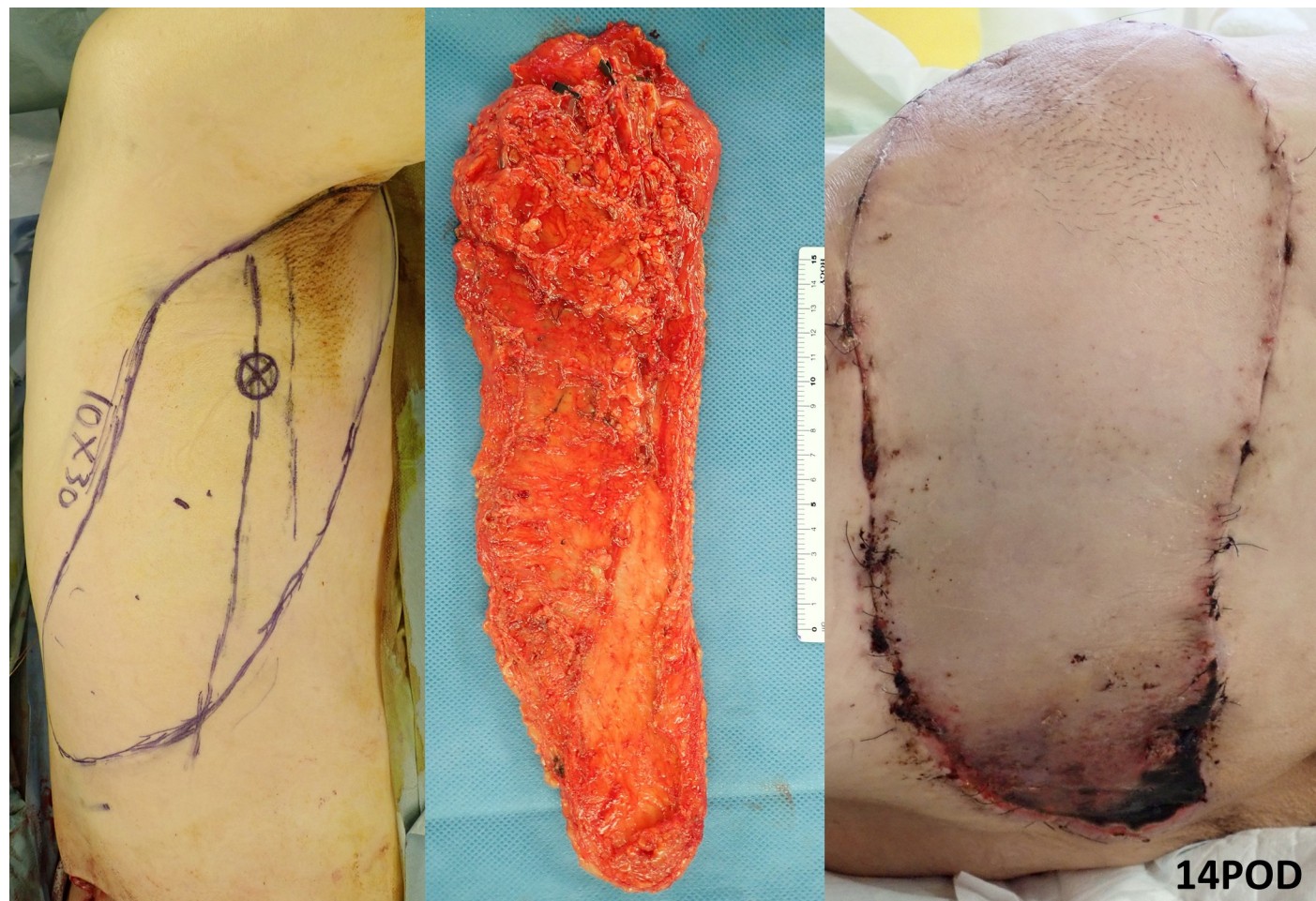

**Fig 9. Case no. 4 in Table 3.** Reconstruction after the extensive resection of the pleomorphic sarcoma with a myxoid change of the buttock. The defect needed a long flap, so that the flap was designed obliquely from the axillary fossa to the central back. The flap was elevated and transferred as a free adipose-cutaneous flap with complete preservation of the LD. Distal flap congestion occurred, resulting in partial necrosis of the distal tip. The distal portion of the flap was considered too medial and out of the perfusion area of the lateral thoracic "WPS" flap. The flap design most likely should have been more parallel to the LD lateral border.

Second, breast DCE-MRI was not originally intended to be utilized for perforator evaluation. However, the area around the LD lateral border was usually included in the normal range of breast cancer evaluation, and a good contrast and visibility of small vessels was obtained because the imaging protocol (eTHRIVE) was optimized for the detection of tumor vascularity. Third, the criteria to be identified as a valid perforator was considered too strict; therefore, small or unclear perforators were ignored. In other words, the present study is practical because only clinically available and usable perforators were accounted for.

## Conclusion

Using the results from DCE-MRI, the present study revealed that there were a substantial number of perforators (TDAP-sc/TDAP-mp/LTAP, total 4.51±1.44) around the LD lateral border. We performed a clinical application of the results for a flap that included all these perforators (WPS) and succeeded in transferring seven large muscle-preserving flaps. The lateral thoracic region has an abundant cutaneous perforator system derived from the TDA and LTA, and the clinical application of the lateral thoracic WPS flap is promising, with a large survival area even with muscle-preserving procedures and requiring only a simple procedure.

## Supporting information

**S1 Fig. Postoperative appearance of the surgical site in Case no. 2 in Table 3 at 4 months after surgery.** The flap survived completely. The donor-site of the split-thickness skin graft have marked pigmentation; however, there was no clinically problematic contracture or motility disturbance.
(TIF)

**S2 Fig. Postoperative appearance of the surgical site in Case no. 5 in Table 3 at 6 months after surgery.** The flap survived completely. The donor-site of primary closure was invisible from the back and there was no clinically problematic contracture or motility disturbance.
(TIF)

## Acknowledgments

We would like to thank Editage (www.editage.com) for English language editing.

Presented at: WSRM2019 (10th Congress of World Society for Reconstructive Microsurgery) in Bologna, Italy.

## Author Contributions

**Conceptualization:** Yu Kagaya, Masaki Arikawa, Takuya Sekiyama, Hideyuki Mitsuwa, Ryo Takanashi, Marie Taga, Satoshi Akazawa, Shimpei Miyamoto.

**Data curation:** Yu Kagaya, Masaki Arikawa, Takuya Sekiyama, Hideyuki Mitsuwa, Ryo Takanashi, Marie Taga, Satoshi Akazawa, Shimpei Miyamoto.

**Formal analysis:** Yu Kagaya.

**Investigation:** Yu Kagaya.

**Methodology:** Yu Kagaya.

**Supervision:** Yu Kagaya, Satoshi Akazawa, Shimpei Miyamoto.

**Validation:** Yu Kagaya, Shimpei Miyamoto.

**Visualization:** Yu Kagaya.

**Writing – original draft:** Yu Kagaya.

**Writing – review & editing:** Yu Kagaya, Masaki Arikawa, Takuya Sekiyama, Hideyuki Mitsuwa, Ryo Takanashi, Marie Taga, Satoshi Akazawa, Shimpei Miyamoto.

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
