## [Decision Letter · Decision Letter 0]

5 Jun 2021

PONE-D-21-06161

The concept of “whole perforator system” in the lateral thoracic region for latissimus dorsi muscle-preserving large flaps: an anatomical study and case series

PLOS ONE

Dear Dr. Kagaya,

Thank you for submitting your manuscript to PLOS ONE. After careful consideration, we feel that it has merit but does not fully meet PLOS ONE’s publication criteria as it currently stands. Therefore, we invite you to submit a revised version of the manuscript that addresses the points raised during the review process.

The manuscript merits consideration and the received reviews support this finding, when submitting your revised version please reply to the issues raised by the reviewers point-by-point. 

I recommend to consider the following in more detail:

- Grading of the postoperative complications by use of a standardised method (see Dindo, Ann Surg, 2004: https://www.assessurgery.com/clavien-dindo-classification/)

- Discuss the morbidity for the donor site in more detail: technical considerations regarding scapular winging associated with mode of dissection, and use of aesthetically unpleasing split-thickness skin grafts for donor site management (number / % of patients?)

- Preoperative imaging: the use of early phase in the modified T1 weighted protocol with contrast agent predominantly reveals the arterial vasculature. How often did the authors observe venous congestion in each group of flaps? Would reviewing the late phase (venous) have aided in selecting the most suitable perforator-pair (artery/vein) candidate? 

We look forward to receiving your revised manuscript.

Kind regards,

David Benjamin Lumenta, MD PhD

Academic Editor

PLOS ONE

Journal Requirements:

PLOS requires an ORCID iD for the corresponding author in Editorial Manager on papers submitted after December 6th, 2016. Please ensure that you have an ORCID iD and that it is validated in Editorial Manager. To do this, go to ‘Update my Information’ (in the upper left-hand corner of the main menu), and click on the Fetch/Validate link next to the ORCID field. This will take you to the ORCID site and allow you to create a new iD or authenticate a pre-existing iD in Editorial Manager. Please see the following video for instructions on linking an ORCID iD to your Editorial Manager account: https://www.youtube.com/watch?v=_xcclfuvtxQ

We ask that you please consider moving Figures 4, 5 ,and 9 to Supporting Information due to their graphic nature. Thank you for your consideration

Reviewers' comments:

Reviewer's Responses to Questions

**Comments to the Author**

1. Is the manuscript technically sound, and do the data support the conclusions?

Reviewer #1: Yes

Reviewer #2: Yes

Reviewer #3: Yes

2. Has the statistical analysis been performed appropriately and rigorously? 

Reviewer #1: Yes

Reviewer #2: Yes

Reviewer #3: Yes

3. Have the authors made all data underlying the findings in their manuscript fully available?

Reviewer #1: Yes

Reviewer #2: Yes

Reviewer #3: Yes

4. Is the manuscript presented in an intelligible fashion and written in standard English?

Reviewer #1: Yes

Reviewer #2: Yes

Reviewer #3: Yes

5. Review Comments to the Author

Reviewer #1: This is a very nicely conducted anatomical study leading to a clinically translatable and elegant concept of the "whole perforator system" or WPS.

This is particularly relevant to spare the LD muscle - which is definitely the direction in which this field should travel. This aspect to preserve muscle function is an important rationale for the study.

The image quality, diagrams of anatomy (figures) and photos are to be commended as they are of high quality and serve very clear illustrations of the concepts and the functional anatomy.

I have detailed in the attached file some issues which should be addressed.

My overall concern is that the use of MRI to define anatomy of the WPS was not then used for the patient series to guide dissection - tailored to each individual patient. Whilst I understand that the dissection using the WPS concept is simplified, the authors themselves state that the dominant systems and communications between the 2 systems (TDA/P and the LTA/P) is highly variable. So I do not know why the authors would not recommend pre-op MRI in patients needing large donor sites harvested from the lateral chest wall. This would certainly still assist with the procedure in knowing which main vascular pedicle is dominant - so to focus on dissecting that (and including the WPS) but also at which level any significant communications occur so that these are not lost or excluded in the dissection.

If these issues can be addressed then I think it is an important piece of work meriting publication.

Reviewer #2: The work presented in this submission represents a clarification of our knowledge but rather than significant new information. The work is presented well and easily read by someone new to this type of work and would allow a new surgeon to understand how they could perform surgery in this area safely. The authors have modified an approach to surgery in this region and provide diagrams to demonstrate how to carry this out safely. Readers who have knowledge of this work will still find the clarification of the vasculature on the lateral chest wall of value. I would recommend acceptance of this submission as it is a progression of knowledge.

The paper does have a significant number of images. the drawn diagrams are particularly useful. The patient images could be published in reduced size if there is limited space.

Reviewer #3: The manuscript “The concept of “whole perforator system” in the lateral thoracic region for latissimus dorsi muscle-preserving large flaps: an anatomical study and case series” is well written and covers an interesting topic.

The perforator sythem of the subscapular artery is generally well studied. Nevertheless, the whole system in this form as presented by the authors is quite interesting and enriches the knowledge in this field.

For this reason, the manuscript may be considered for publication in Plos ONE.

Nevertheless, I have a few questions and suggestions for improvement.

1. Please standardize your complications. For example by means of the Clavien Dindo classification. Please pay special attention to the donor site morbidity, considering the size of the flap.

2. When describing how the flap should be harvested, it is recommended to dissect from the lateral thoracic artery subfacially at the serrates anterior muscle to the thoracodorsal artery. If this is done according to your recommendation, the lateral thoracic nerve cannot be preserved.

Thus, the caudal portion of the serratus muscle would be denervated. Is this the case?

If so, this should be explicitly stated as a limitation of the flap. Even if a winging scapula can be avoided by preserving the upper portion of the muscle, the denervation of most of the serratus anterior muscle is a relevant donor site morbidity and should be noted.

3. Have the preoperatively evaluated perforators and their interopertive localization been confirmed in every case? Please add this to the manuscript.

4. If the donor site is covered with split skin, the donor site morbidity is still not ideal in my opinion compared to a Kiss ALT flap with direct closure, for example. I think this could also be mentioned in the discussion.

Overall, I would still like to congratulate the authors on their good results.

6. PLOS authors have the option to publish the peer review history of their article (what does this mean?). If published, this will include your full peer review and any attached files.

Reviewer #1: **Yes: **Peter A Barry

Reviewer #2: No

Reviewer #3: No

---

## [Author Response · Author response to Decision Letter 0]

22 Jun 2021

Editor:

I recommend to consider the following in more detail:

- Grading of the postoperative complications by use of a standardised method (see Dindo, Ann Surg, 2004: https://www.assessurgery.com/clavien-dindo-classification/)

- Discuss the morbidity for the donor site in more detail: technical considerations regarding scapular winging associated with mode of dissection, and use of aesthetically unpleasing split-thickness skin grafts for donor site management (number / % of patients?)

Response:

We added the ‘postoperative complication section (Clavien-Dindo Classification) in Table 3. Accordingly, the ‘recipient vessel’ section was deleted to simplify the table. 

Dissection under the fascia of serratus anterior muscle can naturally cause long thoracic nerve injury, which can cause denervation of the serratus anterior muscle. However, no clinically problematic signs of shoulder movement disorder occurred in the present cases. The root of the long thoracic nerve is distant from the root of the flap pedicle; thus, the proximal part of the long thoracic nerve was preserved and the denervation of serratus anterior muscle was considered partial. However, surgeons should keep complications in mind when elevating this flap. The above contents were added to the Discussion section. (L295-301)

In terms of aesthetic results of donor-site, primary closure of the donor site can cause breast deformity (lateral traction) (mentioned in the Discussion section). Of course, skin graft scars are not aesthetically favorable; however, the wide LDMC flap also requires skin grafting. The donor scar is considered more inconspicuous from the back in comparison to a traditional LDMC flap. Even the scar of skin graft can be hidden—to some extent—in the back view. The above contents were added to the Discussion section. (L287-289) 

- Preoperative imaging: the use of early phase in the modified T1 weighted protocol with contrast agent predominantly reveals the arterial vasculature. How often did the authors observe venous congestion in each group of flaps? Would reviewing the late phase (venous) have aided in selecting the most suitable perforator-pair (artery/vein) candidate? 

Response:

The possibility of postoperative congestion of the flap is considered very low. The congestion of Case 4 (Fig.9) was derived from a mistake in the flap design, not from the undeveloped perforator vein. However, the above indication is an important and novel point of view. 

Past studies evaluated perforators using contrast-enhanced CT in the early phase (arterial phase)1,2. Naturally perforating arteries and veins exist together; thus, when evaluating perforating arteries, perforating veins will usually be included. However, the LTA and V sometimes exist separately in the proximal region3 , in which late phase DCE-MRI can be useful for distinguishing the LTA and V. Late phase evaluation may be useful for evaluating flap venous perfusion in some cases. The idea is an important suggestion and we want to make it a future challenge.

1. Mun GH, Kim HJ, Cha MK, Kim WY. Impact of perforator mapping using multidetector-row computed tomographic angiography on free thoracodorsal artery perforator flap transfer. Plast Reconstr Surg. 2008 Oct;122(4):1079-88.

2. Saba L, Atzeni M, Ribuffo D, Mallarini G, Suri JS. Analysis of deep inferior epigastric perforator (DIEP) arteries by using MDCTA: Comparison between 2 post-processing techniques. Eur J Radiol. 2012 Aug;81(8):1828-33.

3. Kim JT, Ng SW, Naidu S, Kim JD, Kim YH. Lateral thoracic perforator flap: additional perforator flap option from the lateral thoracic region. J Plast Reconstr Aesthet Surg. 2011 Dec;64(12):1596-602.

Journal Requirements:

We ask that you please consider moving Figures 4, 5 ,and 9 to Supporting Information due to their graphic nature. Thank you for your consideration

Response:

We want to include the case reports in the manuscript because the actual case descriptions and pictures are considered helpful and informative for surgeons. However, it is also true that a large number of pictures was used in comparison to a usual manuscript. Figure 4C, Figure 5C was moved to the Supporting Information section, and Figure 9A/B was integrated and compressed into one file in Figure 9.

Reviewer #1: 

This is a very nicely conducted anatomical study leading to a clinically translatable and elegant concept of the "whole perforator system" or WPS.

This is particularly relevant to spare the LD muscle - which is definitely the direction in which this field should travel. This aspect to preserve muscle function is an important rationale for the study.

The image quality, diagrams of anatomy (figures) and photos are to be commended as they are of high quality and serve very clear illustrations of the concepts and the functional anatomy.

I have detailed in the attached file some issues which should be addressed.

My overall concern is that the use of MRI to define anatomy of the WPS was not then used for the patient series to guide dissection - tailored to each individual patient. Whilst I understand that the dissection using the WPS concept is simplified, the authors themselves state that the dominant systems and communications between the 2 systems (TDA/P and the LTA/P) is highly variable. So I do not know why the authors would not recommend pre-op MRI in patients needing large donor sites harvested from the lateral chest wall. This would certainly still assist with the procedure in knowing which main vascular pedicle is dominant - so to focus on dissecting that (and including the WPS) but also at which level any significant communications occur so that these are not lost or excluded in the dissection.

Response:

Of course, preoperative DCE-MRI is desirable for grasping the vascular anatomy of the target area, however, many obstacles exist in actual clinical settings, including cost, insurance coverage, time limitations in relation to MRI reservation, labor for the patient, and so on. Doppler ultrasonography is useful for a quick and easy evaluation of cutaneous perforators. The accuracy and objectivity are considered inferior to that DCE-MRI; however, that does not matter for surgical procedure because all of the perforators are included in the subcutaneous fat and the individual perforators are not actually detected or dissected in the current technique. Acknowledging the rough location of perforators is sufficient for elevation of the flap. 

In the present study, breast DCE-MRI was originally intended to be utilized for cancer evaluation., but not for perforator evaluation, as mentioned in the Discussion. Thus, DCE-MRI was not associated with the problems of cost or time. It was not possible to detect all communication between the TDA/V and LTA/V, even by the DCE-MRI; however, both visible and invisible communications of the two vascular systems could be included in the lateral thoracic WPS flap (mentioned in the Discussion, L309-311).

If these issues can be addressed then I think it is an important piece of work meriting publication.

Reviewer #2: 

The work presented in this submission represents a clarification of our knowledge but rather than significant new information. The work is presented well and easily read by someone new to this type of work and would allow a new surgeon to understand how they could perform surgery in this area safely. The authors have modified an approach to surgery in this region and provide diagrams to demonstrate how to carry this out safely. Readers who have knowledge of this work will still find the clarification of the vasculature on the lateral chest wall of value. I would recommend acceptance of this submission as it is a progression of knowledge.

The paper does have a significant number of images. the drawn diagrams are particularly useful. The patient images could be published in reduced size if there is limited space.

Response:

We want to include the case reports in the manuscript because the actual case descriptions and pictures are considered helpful and informative for surgeons. However, it is also true that a large number of pictures was used in comparison to a usual manuscript. Figure 4C, Figure 5C was moved to the Supporting Information section, and Figure 9A/B was integrated and compressed into one file in Figure 9.

Reviewer #3: 

The manuscript “The concept of “whole perforator system” in the lateral thoracic region for latissimus dorsi muscle-preserving large flaps: an anatomical study and case series” is well written and covers an interesting topic.

The perforator sythem of the subscapular artery is generally well studied. Nevertheless, the whole system in this form as presented by the authors is quite interesting and enriches the knowledge in this field.

For this reason, the manuscript may be considered for publication in Plos ONE.

Nevertheless, I have a few questions and suggestions for improvement.

1. Please standardize your complications. For example by means of the Clavien Dindo classification. Please pay special attention to the donor site morbidity, considering the size of the flap.

Response:

A Postoperative Complications section (Clavien-Dindo Classification) was added in Table 3, and the descriptions about donor-site morbidity, including serratus anterior muscle denervation, were added to the Discussion. (L295-301)

2. When describing how the flap should be harvested, it is recommended to dissect from the lateral thoracic artery subfacially at the serrates anterior muscle to the thoracodorsal artery. If this is done according to your recommendation, the lateral thoracic nerve cannot be preserved.

Thus, the caudal portion of the serratus muscle would be denervated. Is this the case?

If so, this should be explicitly stated as a limitation of the flap. Even if a winging scapula can be avoided by preserving the upper portion of the muscle, the denervation of most of the serratus anterior muscle is a relevant donor site morbidity and should be noted.

Response:

As above indicated, dissection under the fascia of serratus anterior muscle can naturally cause long thoracic nerve injury, which can cause denervation of the serratus anterior muscle. However, no clinically problematic signs of shoulder movement disorder occurred in the present cases. The root of the long thoracic nerve is distant from the root of the flap pedicle; thus, the proximal part of the long thoracic nerve was preserved and the denervation of serratus anterior muscle was considered partial. However, surgeons should keep complications in mind when elevating this flap. The above contents were added to the Discussion section. (L295-301)

3. Have the preoperatively evaluated perforators and their interopertive localization been confirmed in every case? Please add this to the manuscript.

Response:

Preoperative DCE-MRI is not indispensable; however, acknowledging the location of dominant perforators by Doppler ultrasonography is useful for selecting the flap type (complete muscle-preserving or muscle-sparing). However, the individual perforators are not actually detected or dissected in the current technique of flap elevation. The perforators around the LD lateral border are automatically included in the subcutaneous fat layer of the flap. The above contents were added to the Results section (clinical application). (L200-203)

4. If the donor site is covered with split skin, the donor site morbidity is still not ideal in my opinion compared to a Kiss ALT flap with direct closure, for example. I think this could also be mentioned in the discussion.

Response:

In terms of the aesthetic results of the donor-site, primary closure of the donor site can cause breast deformity (lateral traction) (mentioned in the Discussion). The skin graft scar is of course not aesthetically favorable; however, the wide LDMC flap also requires skin grafting. The donor scar is more inconspicuous from the back than that of traditional LDMC flap. Even if the scar of skin graft can be hidden to some extent in the back view. 

The above-mentioned Kiss ALT flap can achieve both large flap elevation and primary closure of the donor-site; however, the location was different and far from the lateral thoracic region; thus, a direct comparison to the current flap (WPS) is considered unreasonable.

The above contents were mentioned in the Discussion section. (L285-289, 294-295)

Overall, I would still like to congratulate the authors on their good results.

---

## [Decision Letter · Decision Letter 1]

20 Aug 2021

The concept of “whole perforator system” in the lateral thoracic region for latissimus dorsi muscle-preserving large flaps: an anatomical study and case series

PONE-D-21-06161R1

Dear Dr. Kagaya,

We’re pleased to inform you that your manuscript has been judged scientifically suitable for publication and will be formally accepted for publication once it meets all outstanding technical requirements.

Kind regards,

David Benjamin Lumenta, MD PhD

Academic Editor

PLOS ONE

Additional Editor Comments (optional):

Reviewers' comments:

Reviewer's Responses to Questions

**Comments to the Author**

1. If the authors have adequately addressed your comments raised in a previous round of review and you feel that this manuscript is now acceptable for publication, you may indicate that here to bypass the “Comments to the Author” section, enter your conflict of interest statement in the “Confidential to Editor” section, and submit your "Accept" recommendation.

Reviewer #3: All comments have been addressed

2. Is the manuscript technically sound, and do the data support the conclusions?

Reviewer #3: Yes

3. Has the statistical analysis been performed appropriately and rigorously? 

Reviewer #3: I Don't Know

4. Have the authors made all data underlying the findings in their manuscript fully available?

Reviewer #3: Yes

5. Is the manuscript presented in an intelligible fashion and written in standard English?

Reviewer #3: Yes

6. Review Comments to the Author

Reviewer #3: my suggestions for improvement were incorporated into the manuscript. In the current version the manuscript can be considered for publication.

7. PLOS authors have the option to publish the peer review history of their article (what does this mean?). If published, this will include your full peer review and any attached files.

Reviewer #3: No

---

## [Editor Report · Acceptance letter]

24 Aug 2021

PONE-D-21-06161R1 

The concept of “whole perforator system” in the lateral thoracic region for latissimus dorsi muscle-preserving large flaps:an anatomical study and case series 

Dear Dr. Kagaya:

I'm pleased to inform you that your manuscript has been deemed suitable for publication in PLOS ONE. Congratulations! Your manuscript is now with our production department. 

Kind regards, 

on behalf of

Professor David Benjamin Lumenta 

Academic Editor

PLOS ONE